# Cellular hallmarks reveal restricted aerobic metabolism at thermal limits

Aitana Neves, Coralie Busso, Pierre Gönczy*

Swiss Institute of Experimental Cancer Research (ISREC), Swiss Federal Institute of Technology (EFPL), Lausanne, Switzerland

**Abstract** All organisms live within a given thermal range, but little is known about the mechanisms setting the limits of this range. We uncovered cellular features exhibiting signature changes at thermal limits in *Caenorhabditis elegans* embryos. These included changes in embryo size and shape, which were also observed in *Caenorhabditis briggsae*, indicating evolutionary conservation. We hypothesized that such changes could reflect restricted aerobic capacity at thermal limits. Accordingly, we uncovered that relative respiration in *C. elegans* embryos decreases at the thermal limits as compared to within the thermal range. Furthermore, by compromising components of the respiratory chain, we demonstrated that the reliance on aerobic metabolism is reduced at thermal limits. Moreover, embryos thus compromised exhibited signature changes in size and shape already within the thermal range. We conclude that restricted aerobic metabolism at the thermal limits contributes to setting the thermal range in a metazoan organism.

## Introduction

All organisms live within a given thermal range, beyond which growth and fecundity decrease (*Pörtner et al., 2006*). Partly as a result, organisms tend to distribute in the ocean and on land according to latitude as well as depth and altitude, although other elements such as availability of food and light also play a role in shaping preferred habitats (*Pörtner, 2002*; *Pörtner et al., 2006*; *Prasad et al., 2011*). Despite their importance, the mechanisms that set the thermal limits remain incompletely understood.

A mismatch between oxygen supply and demand has been suggested to play a role in setting thermal limits in multicellular organisms. This hypothesis, referred to as the oxygen- and capacity-limited thermal tolerance (OCLTT), derives in part from the observation that oxygen partial pressure in aquatic organisms is constant within a given thermal range and decreases both below the lower thermal limit and above the upper thermal limit (*Pörtner, 2002*; *Pörtner et al., 2006*). In agreement with this hypothesis, the metabolic status of some aquatic organisms has been shown to peak at a given temperature and to decrease both below and above that (*Melzner et al., 2006*; *Wittmann et al., 2008*). Interestingly too, tolerance to high temperatures is increased in an amphibian crab when the animal is in the air compared to when it is in water, reflecting the reduced cost of oxygen supply in air (*Giomi et al., 2014*), again supporting the OCLTT hypothesis. Overall, these data suggest that thermal limits in complex organisms are characterized by a mismatch in oxygen supply and demand, which would result in reduced energy production and thus limit reproduction and growth (*Pörtner, 2002*; *Pörtner et al., 2006*).

Intriguingly, the temperature-dependence of oxygen diffusion is significantly lower than that of metabolism (*Woods, 1999*), raising the question of how oxygen supply and demand can be matched, even within the thermal range. One possible solution is suggested by the observation that body size decreases with augmented temperature in the vast majority of ectotherms ('temperature-size rule') (*Atkinson, 1994*; *Forster et al., 2012*), thereby increasing surface to volume ratio and thus potentially

*For correspondence: pierre. gonczy@epfl.ch

**Reviewing editor**: Fiona M Watt, King's College London, United Kingdom

**eLife digest** An organism can thrive within a certain range of temperatures, beyond which it is less able to grow and reproduce. Different species are adapted to live in environments of different temperatures and this influences where on the planet they can be found.

Researchers have suggested that the optimum temperature range for an organism may be influenced by its oxygen supply. The cells of most organisms need oxygen to produce chemical energy from sugars in a process called respiration. Within their normal temperature range, cells in the body of an organism are adapted to be able to take up sufficient oxygen to produce the energy they need. However, if the temperature rises above or falls below the limits of this range, the uptake of oxygen into cells may work less efficiently.

Neves et al. tested this idea by studying the embryos of two different species of nematode worms grown at the limits of their respective temperature ranges. This led to several changes in the appearance of the embryos. For example, the embryos were larger than normal when grown at the lower end of their temperature ranges, but were smaller when grown at temperatures close to their upper limit. Shape changes were also seen: the embryos of both species were longer when grown at higher temperatures, a change that increases their surface area relative to their volume and may improve their ability to take up oxygen. Further experiments showed that disrupting respiration in the worms could lead to similar size and shape changes within the thermal range.

Neves et al.'s findings provide experimental support that respiration plays an important role in setting the temperature ranges in which organisms can live. The next challenge will be to identify the genes that influence the capacity of respiration in these cells, which may help to explain how particular species have adapted to specific environments.

oxygen availability. In support of this, the slope of this 'temperature-size rule' is steeper for aquatic organisms than terrestrial organisms, in agreement with the lower availability of oxygen in water compared to air (*Forster et al., 2012*). This has led to the suggestion that alterations in cell size in response to changes in temperature within the thermal range are adaptive responses to preserve aerobic capacity, which has been dubbed the MASROS hypothesis (Maintain Aerobic Scope—Regulate Oxygen Supply) (*Atkinson et al., 2006*). What happens beyond the thermal limits within this conceptual framework? One might expect that thermal limits could be characterized by further changes in cell size and potentially also cell shape, in an attempt to increase the available surface area and thus maximize oxygen availability. Furthermore, the MASROS hypothesis predicts that aerobic metabolism, measured as respiration, should decrease beyond both the lower and the upper thermal limit as compared to within the thermal range, once the organism can no longer compensate for the insufficient oxygen availability. To our knowledge, these central predictions of the MASROS hypothesis have not been challenged experimentally in an integrative fashion. Therefore, the extent to which restricted aerobic metabolism is a general principle characterizing thermal limits remains unclear.

## Results and discussion

### Defining thermal limits

We determined embryonic viability in a range of temperatures for *Caenorhabditis elegans* and *Caenorhabditis briggsae* and operationally defined the thermal limits as the upper and lower edges of the temperature range within which >90% of embryos hatched. We thus found that the thermal limits of *C. elegans* were of 12°C and 25°C (*Figure 1A*), and those of *C. briggsae* of 14°C and 27°C (*Figure 1B*), in line with the fact that *C. briggsae* usually lives in warmer climates than *C. elegans* (*Prasad et al., 2011*). The thermal range defined by these upper and lower limits ensures robust propagation of the population and is narrower than merely the reproductive range for *C. elegans* (9°C–26°C [*Anderson et al., 2011*]) or *C. briggsae* (14°C–30°C [*Anderson et al., 2007*; *Prasad et al., 2011*]).

### Cellular features within the thermal range

We reasoned that identifying cellular features that operate differently beyond the thermal limits defined above, as compared to within the thermal range, might reveal critical mechanisms acting at

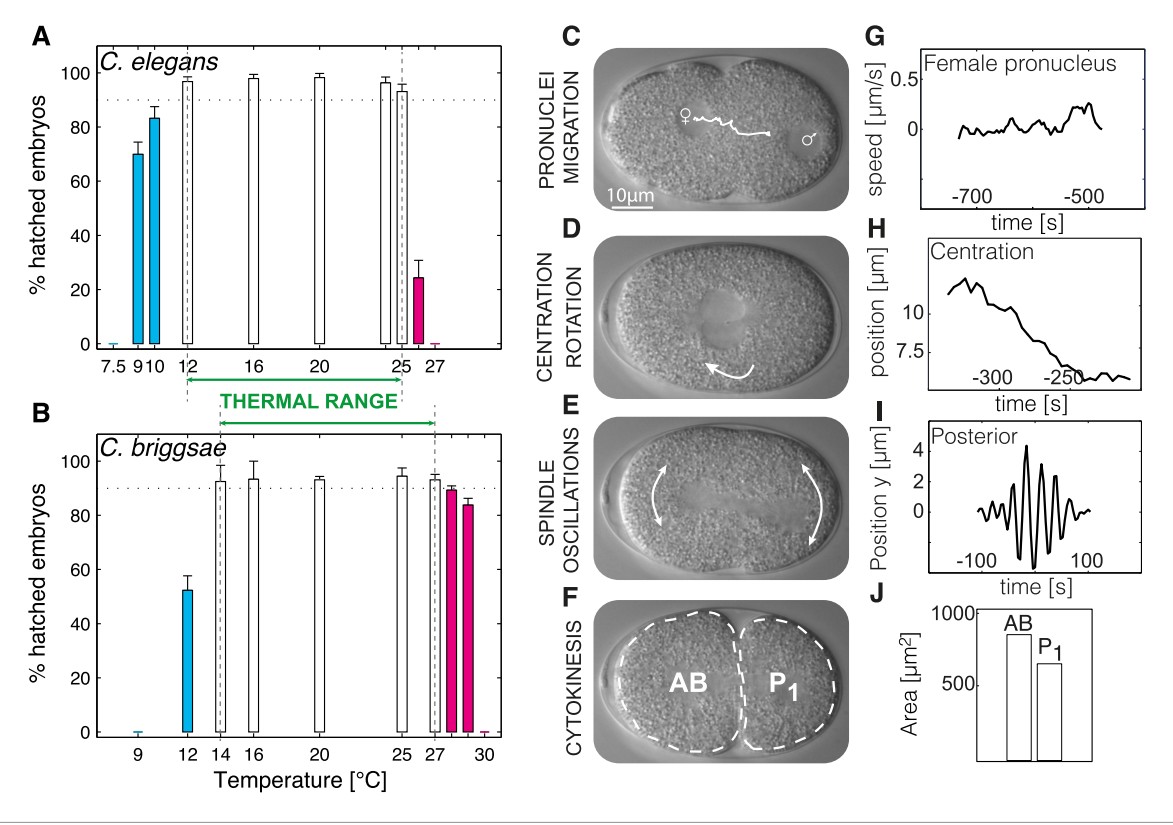

**Figure 1**. Defining the thermal range and quantifications. (**A** and **B**) Progeny tests were performed on acclimated *C. elegans* worms from 7.5°C to 27°C and *C. briggsae* worms from 9°C to 30°C. Dotted line highlights 90% embryonic viability. Temperatures below 20°C exhibiting less than 90% viability are shown in cyan, temperatures above 20°C exhibiting less than 90% viability in magenta. Between panels **A** and **B**, we show the thermal range of each species. Error bars show SEM. (**C–F**) Stills from a time-lapse temperature-controlled DIC microscopy recording of a first-cell stage embryo at the indicated stages (**G–J**) Examples of feature quantification at the different cellular stages (24°C): female pronucleus speed (**G**), pronuclei position during centration-rotation (**H**), spindle pole oscillations (**I**), as well as areas of the AB (anterior) and $P_1$ (posterior) daughter cells (**J**). See 'Materials and methods' for details on the quantifications. *Figure 1—figure supplement 1* shows the temperature control setup. *Figure 1—source data 1* lists all the quantified features and their thermal response within and beyond the thermal range.

The following source data and figure supplements are available for figure 1:

**Source data 1**. Quantified features.

**Figure supplement 1**. Temperature-control setup.

these limits. In order to systematically identify such limit-sensitive features, we first analyzed cellular processes within the thermal range. We conducted this analysis initially in *C. elegans* embryos, but then also studied embryos of *C. briggsae*, which has been estimated to have diverged from *C. elegans* 18–100 million years ago (*Stein et al., 2003*; *Cutter, 2008*), thus probing evolutionary conservation of putative limit-sensitive features. Using temperature-controlled time-lapse DIC (Differential Interference Contrast) microscopy and semi-automated quantifications of the resulting movies with in-house scripts (*Figure 1C–J*, *Figure 1—figure supplement 1*, 'Materials and methods', and *Source code 1*), we measured 35 cellular features that describe the main events of the first cell cycle of *C. elegans* embryos (*Figure 1C–F*). In brief, after fertilization, the female pronucleus migrates towards the male pronucleus (*Figure 1C*). After their meeting, the pronuclei move to the embryo center whilst undergoing a 90°C rotation (*Figure 1D*). The nuclear envelopes then break down, followed by assembly of the mitotic spindle, which moves slightly to the posterior during the remainder of mitosis whilst oscillating perpendicular to the anterior–posterior axis (*Figure 1E*). This results in the asymmetric division of the one-cell stage embryo into a larger anterior cell and a smaller posterior one (*Figure 1F*). Our analysis established that the vast majority of the monitored

features were temperature-dependent within the thermal range (*Figure 1—source data 1*). Interestingly, some features, including the fraction of time spent in mitosis (*Figure 2B*) and cell division asymmetry (*Figure 2C*), exhibited a temperature-independent behavior, suggesting that temperature-compensation mechanisms are also at play.

## Mitosis duration and cell division asymmetry are sensitive to the thermal limits

We then were in a position to identify cellular features that might operate differently beyond the thermal limits compared to within the thermal range. We found that although some features exhibited the same thermal response as within the thermal range, others responded differently, suggesting that they were sensitive to the thermal limits (*Figure 1—source data 1*). Thus, the duration of mitosis, which decreased monotonically with increasing temperatures within the thermal range, plateaued beyond both lower and upper thermal limits in *C. elegans* (*Figure 2A*). Moreover, although *C. briggsae* can develop at warmer temperatures than *C. elegans* (*Figure 1A–B*) (*Prasad et al., 2011*), we found that cell cycle duration was not faster in *C. briggsae* than in *C. elegans* at any temperature (*Figure 3A*, compare with *Figure 2A*). Interestingly, cell cycle duration within the thermal range was well described by Arrhenius kinetics in *C. elegans* (92% of explained variance; 'Materials and methods') (*Arrhenius, 1915*). In *C. briggsae*, by contrast, the data beyond 25℃ reduced the explained variance from 86% to 39%, suggesting that cell cycle duration plateaued already below the upper thermal limit in this species, underscoring the fact that mitosis duration is a limit-sensitive feature.

We also observed that the asymmetry of the first cell division in *C. elegans*, which was constant within the thermal range, decreased below 12℃ (*F*-test p = 0.0005) and increased above 25℃ (*F*-test p < 10$^{-7}$) (*Figure 2C*; see 'Materials and methods' for statistics). In *C. briggsae*, the asymmetry of the first cell division also increased beyond the upper thermal limit, at both 28℃ and 29℃ (*F*-test p-value < 10$^{-10}$) (*Figure 3C*). However, a reciprocal decrease was not observed at the lower thermal limit in this species, perhaps because spindle pole oscillations are weaker in *C. briggsae* than in *C. elegans* (*Riche et al., 2013*) (compare panel D in *Figures 2, 3*), potentially limiting the dynamic range over which asymmetry can be tuned.

## Embryo size and shape are sensitive to the thermal limits

Our analysis also revealed interesting alterations in embryo geometry at the upper and at the lower thermal limits. Thus, embryo size was larger at the lower end of the thermal range (i.e., 12℃ for *C. elegans*, *Figure 2E*; 14℃ for *C. briggsae*, *Figure 3E*), and tended to decrease with increasing temperature within the thermal range. This is in line with the 'temperature-size rule' observed in the vast majority of ectotherms (*Atkinson, 1994*; *Forster et al., 2012*), and in agreement with previously reported data for *C. elegans* at 10℃ vs 20℃ (*Van Voorhies, 1996*). Strikingly, below the lower thermal limit, embryo size was actually significantly reduced in both *C. elegans* and *C. briggsae* (*Figures 2E, 3E*; *F*-test p < 10$^{-7}$ and p = 0.001, respectively). Such a reversal of the temperature size rule below the lower thermal limit has also been reported in protists and in *Drosophila* (*Karan et al., 1998*; *Atkinson et al., 2003*). These observations are compatible with the MASROS hypothesis, which posits that such a size decrease below the lower thermal limit may reflect cold-inhibited mitochondrial function (*Atkinson et al., 2006*).

Beyond the upper thermal limit, we observed a plateau in the size of both *C. elegans* and *C. briggsae* embryos (*Figures 2E–3E*). Interestingly, however, we observed that embryos in both species were more elongated beyond the upper thermal limit (*Figures 2F–3F*; *F*-test p = 0.0004 and *F*-test p < 10$^{-10}$, respectively). Such an elongation results in an increase of the surface area, thus potentially augmenting its availability for oxygen diffusion (*Figure 2—figure supplement 1*). Overall, these results reveal that changes in cell size and shape are signature hallmarks of the thermal limits.

## Cellular hallmarks of the thermal limits are recapitulated when impairing aerobic metabolism

Do the observed changes in embryo size below the lower thermal limit and of shape above the upper thermal limit reflect an adaptation to reduced aerobic metabolism at those temperatures? We set out to explore this possibility by determining the extent of respiration at different temperatures in wild-type *C. elegans* embryos. As shown in *Figure 4A*, we found that respiration increased exponentially within the thermal range, as predicted by Arrhenius-like kinetics

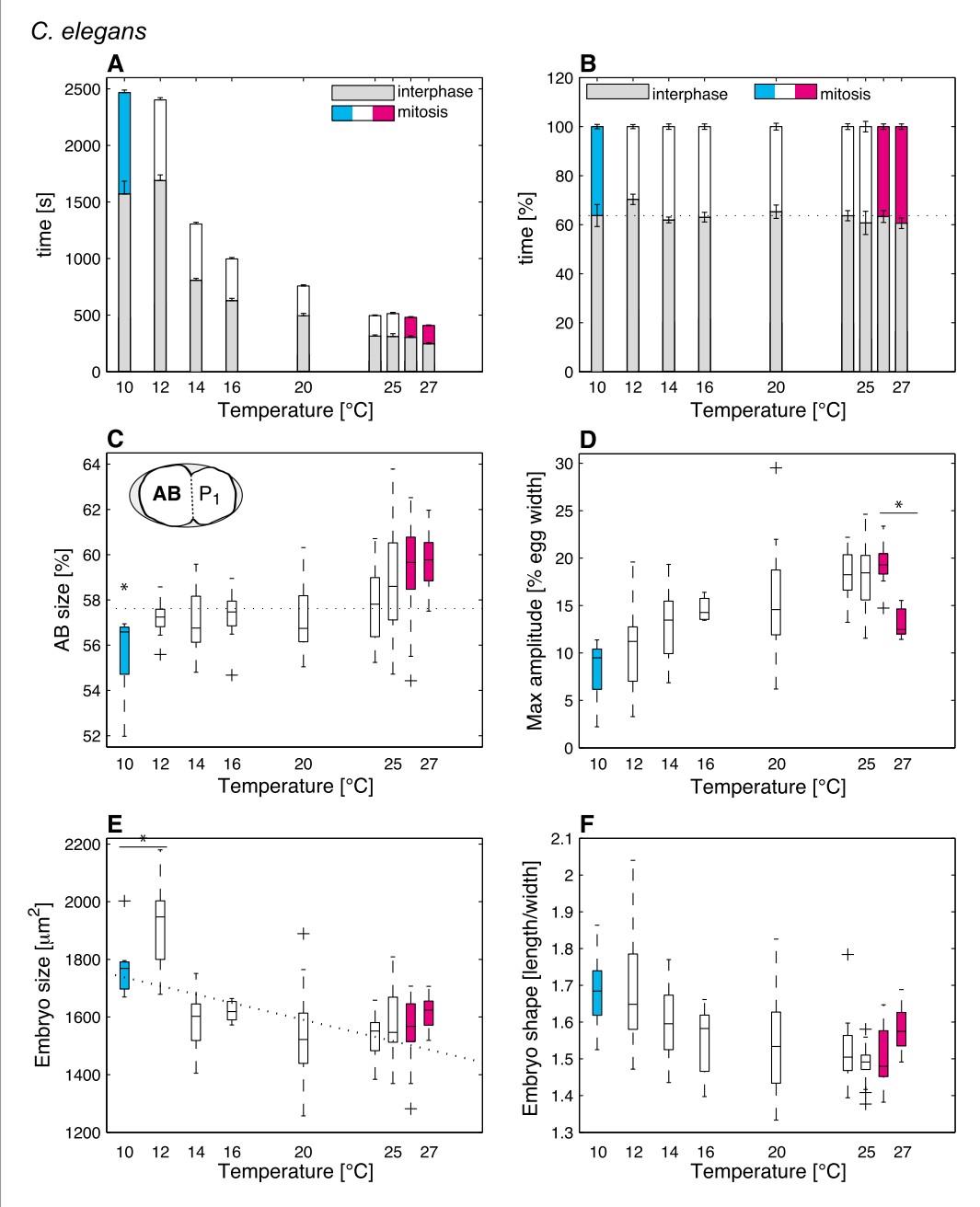

**Figure 2**. *C. elegans* thermal responses. (**A**) Cell cycle duration as a function of temperature (error bars show SEM). (**B**) Relative cell cycle duration as a function of temperature (error bars show SEM). (**C**) Relative size of the AB blastomere as a function of temperature. Dotted line represents the average relative size within the thermal range (57.4%). (**D**) Maximum amplitude of posterior pole oscillations during anaphase. (**E**) Embryo size as a function of temperature. Dotted line shows a linear regression of the data within the thermal range (white boxes). (**F**) Embryo shape, measured as the ratio of embryo length over embryo width, as a function of temperature. See main text for p-values. Color code (for the whole figure): white bars show data within the thermal range. Colored bars show data below (cyan) and above (magenta) the thermal limit. Boxplots show median as well as 25th and 75th percentiles. Whiskers extend to the most extreme points not considered outliers (i.e., within 99.3% coverage). Note that the variance of cellular features does not increase beyond the thermal limits as compared to within the thermal range. *Figure 2—figure supplement 1* depicts embryo size and shape at various temperatures.

The following figure supplement is available for figure 2:

*Figure 2. continued on next page*

*Figure 2. Continued*

**Figure supplement 1**. Embryo size and shape at various temperatures, exaggerating the actual differences for visualization purposes.

(*Arrhenius, 1915*). Strikingly in addition, this analysis uncovered that respiration departed from Arrhenius-like kinetics both below the lower thermal limit (*F*-test p-value $< 10^{-4}$) and above the upper thermal limit (*F*-test p-value $< 10^{-10}$), in support of reduced respiration at those temperatures (*Figure 4A*). Although we do not know whether the observed relative reduction in aerobic capacity beyond both thermal limits as compared to within the thermal range contributes to increased lethality at those limits, our results show a clear correlation between these features.

One possibility to interpret these data is that the energetic needs of the embryo are not satisfied beyond the thermal limits due to insufficient aerobic metabolism. Another possibility is that these needs are actually fulfilled to some extent despite reduced respiration, either because other metabolic routes are used to a larger relative extent or because embryos are metabolically depressed at the thermal limits and thus require less energy altogether. We reasoned that if aerobic metabolism became insufficient beyond the thermal limits, then further compromising mitochondrial activity should have more of an impact at the thermal limits than within the thermal range. By contrast, if energetic needs could be fulfilled at the least to some extent despite reduced respiration beyond the thermal limits, then further compromising mitochondrial activity should have less of an impact at the thermal limits than within the thermal range. Therefore, to distinguish between these two possibilities, we depleted three components of the mitochondrial respiratory chain using RNAi: the beta-subunit of ATP synthase ATP-2 (*Tsang et al., 2001*), a complex V component, the subunit of the mitochondrial complex I NUO-1 (*Tsang et al., 2001*), and the component of the mitochondrial complex III CYC-1 (*Dillin et al., 2002*). We ascertained that embryonic respiration was reduced in *cyc-1(RNAi)* embryos, reaching on average 56% $\pm$ 13% of the wild-type levels under the assay conditions (t-test p-value $< 10^{-3}$; see 'Materials and methods'). Since ATP-2 and NUO-1 are part of complex V and I, respectively, respiration may still occur upon their depletion, even if the mitochondrial respiratory chain is compromised (*Braeckman et al., 2009*), so that respiration measurements may not have been telling in these cases. Importantly, we found that all three RNAi conditions were embryonic lethal to some extent (*Figure 4B–D*), probably owing to decreased energy production through respiration, although we cannot exclude that the observed lethality stems from changes in pH or increased reactive oxygen species. Importantly, in addition, this analysis uncovered that embryonic lethality was reduced towards the lower thermal limit as compared to within the thermal range in all three cases, as well as towards the upper thermal limit in both *cyc-1(RNAi)* and *nuo-1(RNAi)* (*Figure 4B–D*). These results offer strong experimental support to the notion that the capacity of the mitochondrial respiratory chain is restricted beyond both thermal limits, and raise the possibility that other metabolic routes are used to a larger relative extent at those temperatures in the face of reduced respiration.

Following up on this result, we set out to test whether the changes in size or shape observed beyond the thermal limits in the wild-type reflect an adaptation response to restricted aerobic capacity. We reasoned that if this were the case, then such changes should occur already within the thermal range of embryos in which components of the mitochondrial respiratory chain are compromised. Interestingly, we found that whereas embryo size was not significantly affected upon RNAi-mediated depletion of *atp-2* (*Figure 4E*), these embryos were more elongated at both 12°C and 16°C (*Figure 4F*, U-test p(12°C) = 0.0034, p(16°C) = 0.0039). In *nuo-1(RNAi)*, embryo size was significantly reduced at both temperatures (U-test p(12°C) = 0.02, U-test p(16°C) $< 10^{-3}$, *Figure 4E*), whereas a similar response was observed in *cyc-1(RNAi)* embryos at 12°C (U-test p(12°C) $< 10^{-3}$, *Figure 4E*). While it remains to be investigated why the cellular consequences of depleting these three components differ to some extent, remarkably, they share the net result of increasing surface to volume ratio within the thermal range, thus mimicking the situation in the wild-type beyond the thermal limits. Therefore, these results strongly support the notion that the uncovered cellular hallmarks observed at the thermal limits of wild-type embryos reflect restricted aerobic capacity.

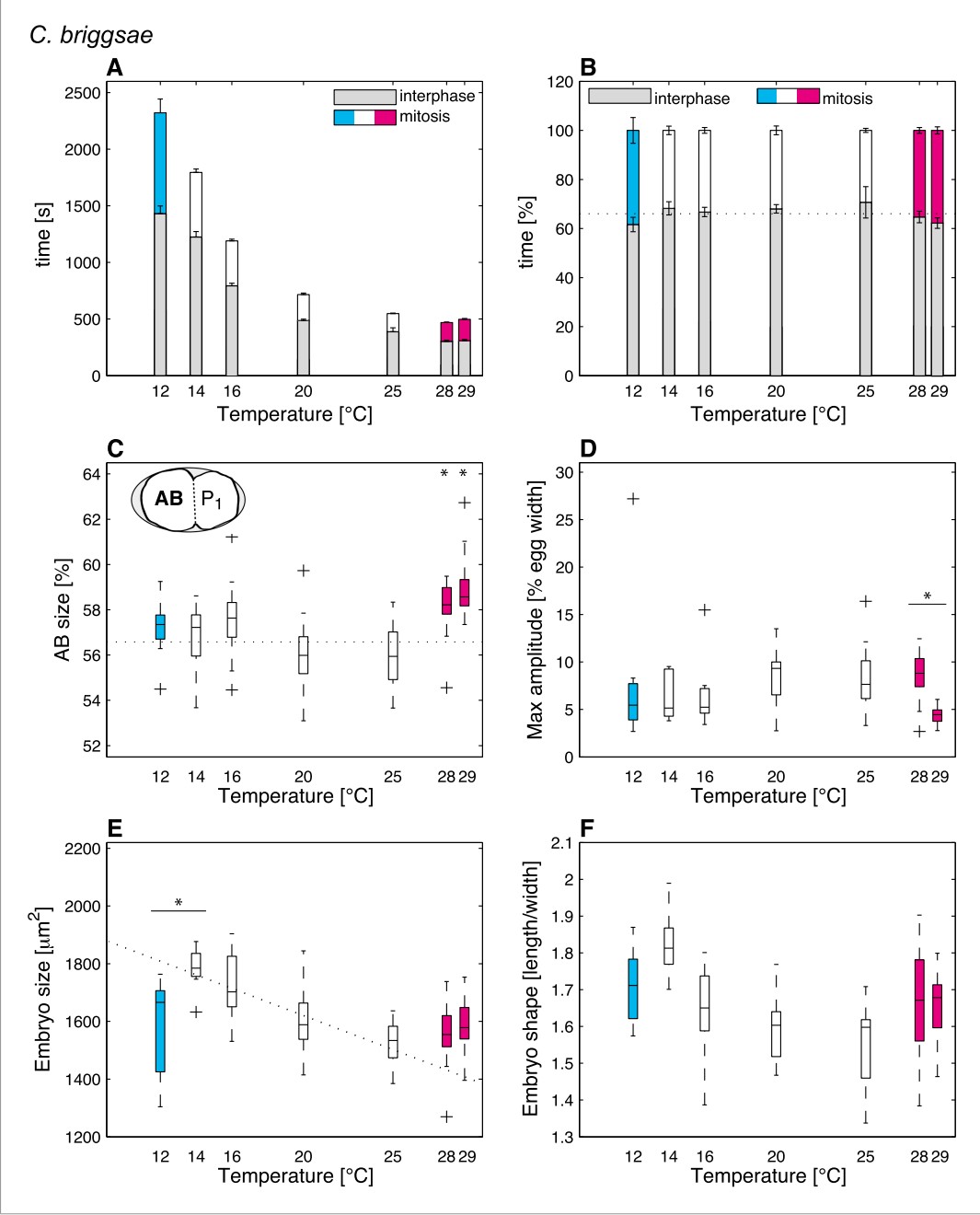

**Figure 3**. Thermal responses in *C. briggsae*. See legend of *Figure 2* and main text for p-values.

## Conclusions

In this work, we assessed the thermal response of cellular features during the first cell cycle of *C. elegans* and *C. briggsae* embryos. Interestingly, we uncovered that the thermal response of select cellular features changed precisely at the limit temperatures defined by embryonic viability tests (see *Figure 1A–B*). While we do not know whether these cellular hallmarks are responsible for the observed increased lethality beyond the thermal limits, we note that a mere 10% decrease in embryonic viability is associated with readily observable cellular changes during the first cell cycle. Importantly, experiments in which mitochondrial respiration is compromised revealed that aerobic metabolism plays a smaller relative role towards the thermal limits than within the thermal range, raising the possibility that other metabolic routes are favored to produce energy.

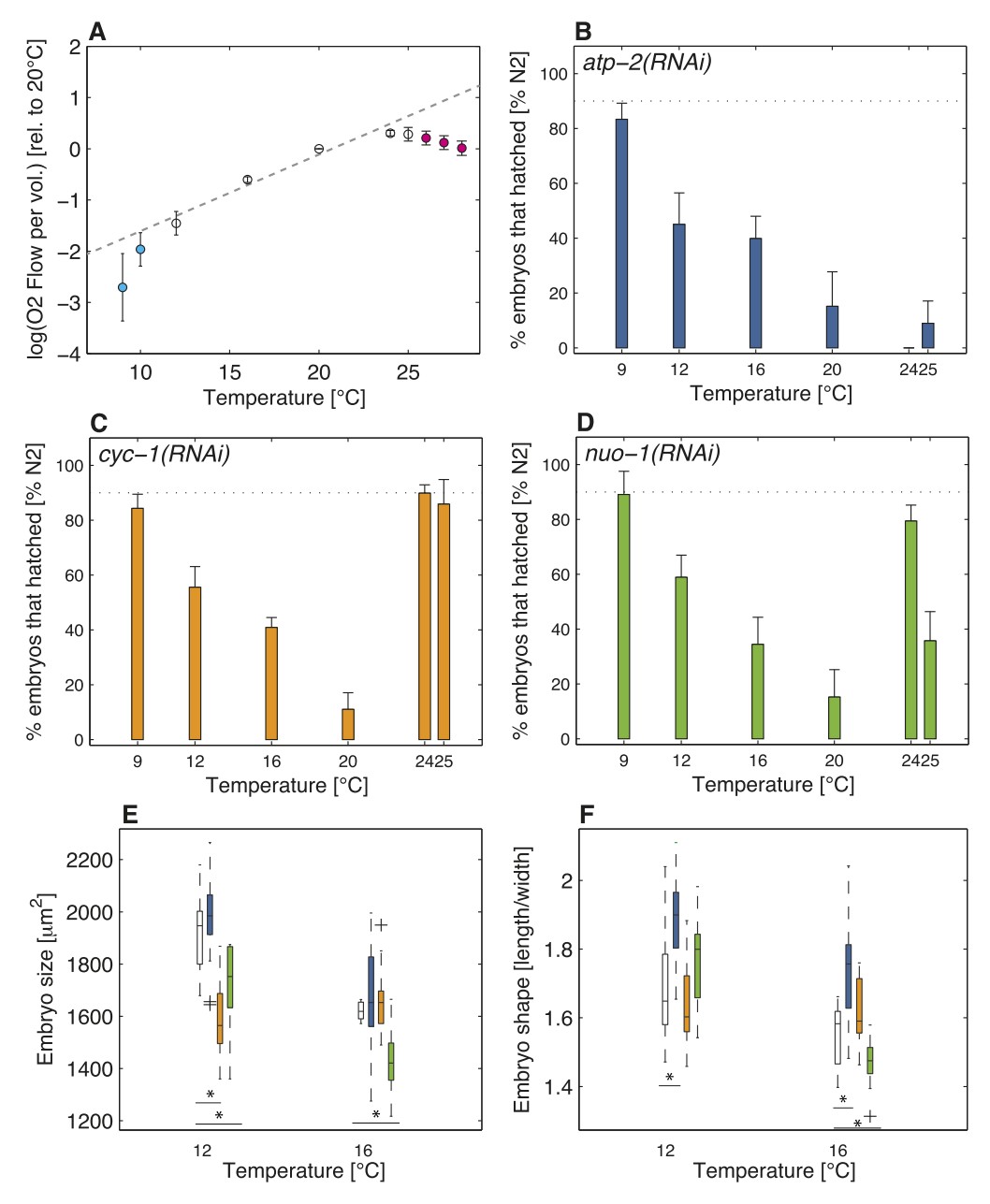

**Figure 4**. Restricted aerobic metabolism at the thermal limits. (**A**) Oxygen consumption (y-axis displays the logarithm of $O_2$ flow per volume) in embryos at different temperatures from 9°C to 28°C. Pooled data from two biological replicates, each with two technical replicates (see 'Materials and methods'). Error bars represent the SEM. Note that respiration increases exponentially between 12°C and 24°C (white discs), as shown by the linear increase in log-scale (gray dashed line shows exponential fit between 12°C and 24°C). Note also that respiration decreases beyond both thermal limits (cyan and magenta discs, respectively), and no longer follows the exponential trend observed within the thermal range. (**B**) Color-code for panels (**B**–**F**): white (wild-type), blue (*atp-2(RNAi)*), orange (*cyc-1(RNAi)*), green (*nuo-1(RNAi)*). Progeny tests on *atp-2(RNAi)* embryos. (**C**) Same as B for *cyc-1(RNAi)*. (**D**) Same as B for *nuo-1(RNAi)*. (**E**) Embryo size as a function of temperature. (**F**) Embryo shape as a function of temperature. See main text for p-values. *Figure 4—figure supplement 1* shows the RNAi feeding times as a function of temperature.

The following figure supplements are available for figure 4:

**Figure supplement 1**. RNAi feeding time as a function of temperature.

**Figure supplement 2**. Progeny tests in *air-1(RNAi)*.

Furthermore, these experiments uncover that the changes in size and shape observed beyond the thermal limits in the wild-type can be recapitulated within the thermal range by impairing aerobic metabolism, strongly supporting the view that these changes arise in response to restricted aerobic metabolism. Together, our work provides critical experimental evidence supporting the notion that restricted aerobic metabolism is a general principle characterizing thermal limits in multicellular organisms in water and on land. Other elements contribute to setting boundary conditions within which a thermal range can be envisaged. Thus, cold-induced increase in unsaturated fatty acids in cyanobacteria, *Arabidopsis* (*Hazel, 1995*) and *C. elegans* contributes to setting the lower thermal limit (*Svensk et al., 2013*), although it only accounts for 16% of the observed difference in cold tolerance at 10°C vs 25°C in *C. elegans* (*Murray et al., 2007*). Moreover, warm-induced increase in post-translational glycosylation also contributes to setting the upper thermal limit in *Drosophila melanogaster*, *Danio rerio* and *C. elegans* (*Radermacher et al., 2014*). In addition, defects in synaptonemal complex assembly (*Bilgir et al., 2013*) and in sperm (*Harvey and Viney, 2007*) contribute to setting the upper organismal limit in *C. elegans*. The restricted aerobic metabolism experimentally uncovered here is another important piece of the puzzle that contributes to defining both thermal limits.

## Note added in proof

Another study investigating the temperature dependence of cell division processes in *C. elegans* and *C. briggsae* was published whilst the present manuscript was under consideration (*Begasse et al., 2015*).

## Materials and methods

### Culture and imaging

All the strains were maintained according to standard procedures (*Brenner, 1974*) in incubators set at the temperature at which embryos would then be imaged. Note, however, that since *C. elegans* was not fully viable above 25°C (see *Figure 1A*), worms were kept at the imaging temperature for only 6–24 hr prior to imaging. Embryos were dissected in 1× M9 medium tempered at the culture temperature, mounted on slides, placed under a coverslip and imaged using time-lapse DIC microscopy. Considering the crowded compressive environment of the uterus in the intact animal, and considering furthermore that the same mounting procedure was followed for all specimens at all temperatures, we surmise that the observed alterations in thermal response of embryo size and shape at given temperatures are not due to the mounting procedure. However, we cannot totally exclude that the observed changes in embryo size and shape may result from differential resilience to pressure of the cover slip used for imaging at the various temperatures.

The recording rate was adjusted as follows (we also mention the number $n$ of embryos that were imaged from each condition):

*C. elegans* (N2): 10°C (9 s, $n = 9$), 12°C (8 s, $n = 12$), 14°C (6 s, $n = 11$), 16°C (6 s, $n = 9$), 20°C (5 s, $n = 20$), 24°C (4 s, $n = 19$), 25°C (4 s, $n = 15$), 26°C (4 s, $n = 16$), 27°C (2 s, $n = 10$).

*C. briggsae* (AF16): 12°C (8 s, $n = 8$), 14°C (7 s, $n = 9$), 16°C (6 s, $n = 16$), 20°C (4.5 s, $n = 21$), 25°C (3 s, $n = 14$), 28°C (2 s, $n = 16$), 29°C (1.5 s, $n = 15$).

*atp-2(RNAi)*: 12°C ($n = 14$), 16°C ($n = 16$). In this condition, only few embryos were imaged over the whole first cell cycle ($n(12°C) = 3$, $n(16°C) = 6$).

*cyc-1(RNAi)*: 12°C ($n = 15$), 16°C ($n = 14$). In this condition, only few embryos were imaged over the whole first cell cycle ($n(12°C) = 2$, $n(16°C) = 4$).

*nuo-1(RNAi)*: 12°C ($n = 8$), 16°C ($n = 15$). In this condition, only few embryos were imaged over the whole first cell cycle ($n(12°C) = 1$, $n(16°C) = 4$).

While imaging, the temperature was regulated by an air-blower that cooled/heated both sample and objective, and which was feedback-controlled by a thermocouple (LABFACILITY ZO-PFA-K-1) inserted next to the embryo (*Figure 1—figure supplement 1A*). We also ensured that the device was well calibrated in the experimental thermal range [8, 32]°C (*Figure 1—figure supplement 1B*).

### Quantifications

Prior to imaging, we made sure that embryos did not touch each other in order to facilitate segmentation. All DIC recordings were analyzed in a semi-automated fashion using Matlab. The analysis pipeline consisted of the following steps:

1. We automatically segmented the eggshell contour using ASSET (*Blanchoud et al., 2010*). All the measured positions were then automatically corrected at each time frame by the centroid of the egg in the same frame. This was an important step because the air-blow from the temperature controller displaced the embryos during the recordings.
2. We detected by careful visual inspection the onset of pseudo-cleavage (deepest furrow), mitotic entry (nuclear envelope breakdown) and cytokinesis (start of membrane invagination). Cytokinesis onset defined time 0; hence, all the times in our analysis were negative.
3. We automatically detected the migrating pronuclei using a custom segmentation algorithm based on (*Hamahashi et al., 2005*). The exact timing of pronuclear meeting was then corrected by manual inspection. The speed of the female pronucleus was computed using its movement along the x-axis.
4. After pronuclear meeting, the spindle poles were manually tracked until completion of centration-rotation. The angular and spatial trajectories were then fitted with the following model:

$$x = \frac{A \cdot |t|^n}{K^n + |t|^n} + cte,$$

where $x$ is the mid-position of the spindle poles or the angle they make with respect to the A-P axis. $K$ represents the time at which centration (resp. rotation) is midway to completion. $A$ relates to the initial position (resp. angle). $cte$ is an offset and $n$ relates to the steepness of the profile. The velocity can then be computed using $v = dx/dt$.
5. After mitotic entry, the spindle poles were manually tracked until oscillations had dampened out. The position along the x-axis was used to compute spindle pole elongation speed towards the anterior and posterior poles, while positions along the y-axis monitored spindle oscillations. In order to retrieve the oscillation frequency, amplitude and duration, we first identified the dominant angular frequency $\omega$ by fast Fourier transform. We then applied a low-pass filter with threshold 3 $2\pi$ $\omega$, to remove the noise, followed by a high-pass filter to remove any drift of the oscillations (with threshold 0.5 $2\pi$ $\omega$). Note that these filters did not change the dominant frequency of the signal, but were useful to better detect the peaks and measure the amplitude of the oscillations. Since the oscillations envelope was not always well fit by a sinusoidal function, we determined the duration of the oscillations by manual inspection of the oscillations profile (after filtering).
6. In order to determine embryo size, the embryo was manually contoured just before cytokinesis onset in order to extract its area.
7. The area of each daughter cell was manually contoured at time $t \cong 0.25 \cdot t_{PM}$, where $t_{PM}$ is the duration of the first cell cycle, defined as the time from pronuclear meeting to cytokinesis onset.

## Progeny tests

In order to perform progeny tests, five to ten young adults were placed on a plate with a 5 μl drop of OP50 and left to lay eggs at the temperature of interest (at least in triplicates). After 2–4 hr, we removed all the adults and counted the number of embryos on the plate (generally between 30 and 100 embryos, except at extreme temperatures beyond the thermal limits where few or no embryos were laid). After a few days at the temperature of interest, we assessed the number of larvae that had hatched.

## Measurement of embryonic respiration

Unsynchronized embryos were obtained by bleaching adult wild-type *C. elegans* worms. The number of embryos per μl was then assessed by optical density ($OD_{595\ nm}$). We measured the respiration of wild-type *C. elegans* embryos from 9°C to 28°C using the Oroboros Oxygraph-2k, following the manufacturer's instructions. Prior to the experiment, a calibration was performed with 1× M9 buffer at 20°C in each chamber. We then dispensed 100,000 embryos in four chambers containing M9 buffer (i.e., 25,000 embryos/chamber): two chambers were used to go down in temperature from 20°C to 9°C, and two chambers were used to go up from 20°C to 28°C. The data from each chamber was normalized to its respiration rate at 20°C. We also repeated the same experiment using 35,000 embryos per chamber (i.e., 140,000 embryos in total).

In order to measure respiration at 20°C upon CYC-1 depletion, we dispensed 2000 wild-type embryos/plate on 16 large Petri dishes with OP50 bacteria as food source. After 28 hr at 20°C, all the resulting larvae were collected by centrifugation and washed three times to remove the OP50. Half of the collected larvae was re-suspended and distributed in 16 large OP50 plates, the other half in

16 large *cyc-1(RNAi)* IPTG feeding plates (prepared the day before and left at room temperature). After 44 hr at 20°C, embryos were collected in both control and *cyc-1(RNAi)* conditions by bleaching adult worms. Respiration was measured in two chambers as follows: after calibrating the machine with 1× M9 buffer at 20°C, 35,000 wild-type embryos were dispensed in each chamber and their respiration measured at that temperature. The chamber was then washed, and we dispensed 35,000 embryos from the *cyc-1(RNAi)* condition and likewise measured their respiration. For each chamber, we compared the respiration of *cyc-1(RNAi)* embryos over wild-type. The experiment was repeated once using 35,000 embryos in the four chambers. Note that the lethality incurred following *cyc-1(RNAi)* in these experiments was less pronounced than that reported in *Figure 4C*, probably owing to the need to scale up to assess respiration in a large number of embryos, such that the reported diminution of respiration is likely an underestimate of the actual impact.

## RNAi

The *C. elegans* ORFeome RNAi library was a gift from Jean-François Rual and Marc Vidal, Harvard Medical School, Boston, USA (*Rual et al., 2004*). Bacterial RNAi feeding strains were prepared as described (*Kamath et al., 2001*). RNAi was performed by feeding early L3 larvae at temperature $T$ with bacteria expressing dsRNA against the target gene for $N$ hours at temperature $T$. The required feeding durations at each temperature were determined by fitting the duration of embryogenesis at 16°C (29 hr), 20°C (18 hr) and 25°C (14 hr) (*Epstein and Shakes, 1995*) with the following equation (*Gillooly et al., 2002*):

$$t(T) = A/\exp\left(\alpha \cdot \frac{T_c}{1 + \frac{T_c}{T_0}}\right),$$

where $T_0$ = 273 K and $T_c$ is the temperature in °C, yielding $\alpha$ = 0.1, in agreement with (*Gillooly et al., 2002*) (*Figure 4—figure supplement 1*). We therefore used this value of $\alpha$ to fit the reported feeding durations from the literature (~72 hr at 15°C [*Ahringer, 2006*], ~47 hr at 20°C [*Afshar et al., 2005*] and ~38 hr at 22°C [*Ahringer, 2006*]), yielding the following feeding durations: 12°C (90 hr), 16°C (65 hr), 20°C (44 hr), 24°C (31 hr).

In order to verify that the results we uncovered in *Figure 4B–D* did not result from a general temperature-dependency in the effectiveness of the RNAi response, we also performed RNAi directed against AIR-1, a serine/threonine kinase required for spindle assembly (*Hannak et al., 2001*), a process not known to be related to metabolic status (*Figure 4—figure supplement 2*).

## Statistical significance of thermal responses

Thermal response within the thermal range was assessed by Pearson correlation. A feature was considered to be temperature-dependent within the thermal range if its Pearson p-value was below 0.0014 (which assumes a Bonferroni correction for multiple-testing 35 features, i.e., 0.05/35 = 0.0014; c.f. *Figure 1—source data 1*).

Beyond the thermal range (TR), we assessed if there was a significant change in the thermal response of the features by $F$-test (nested-model analysis). For temperature-independent features, our first model was a simple regression $y = mean$ (feature within TR). This model was nested within our second model $y = mean$ (feature within TR) + $\beta(T - 25)$ ($T \geq 25$°C) which accounted for the potential change in the feature's thermal response beyond 25°C (a similar model was implemented for *C. elegans* lower thermal limit, as well as for *C. briggsae* at its respective thermal limits). We then determined if the second model (which has more parameters and therefore always fits the data better) *significantly* improved the fit of the data using an $F$-test. The $F$ statistic is given by:

$$F = \frac{\left(\frac{RSS_1 - RSS_2}{p_2 - p_1}\right)}{\left(\frac{RSS_2}{n - p_2}\right)},$$

where $RSS_i$ is the residual sum of squares of model $i$, $p_i$ the number of parameters of model $i$ and $n$ the number of data points. Under the null hypothesis, $F$ follows an $F$-distribution with ($p_2 - p_1$; $n - p_2$) degrees of freedom.

For temperature-dependent features, the first model was an exponential fit of the data within the thermal range: for example, for *C. elegans*, $y = y_0 \cdot \exp(\alpha \cdot T (T \in [12°, 25°]))$. The nested model included a linear regression for the data above 25°C (or similar for *C. elegans* lower thermal limit, as well as for *C. briggsae* at its respective thermal limits):

$$y = y_0 \cdot \exp(\alpha \cdot T(T \in [12°, 25°])) + \beta(T - 25)(T \geq 25°C).$$

In some temperature-dependent cases, the data within the thermal range was better fitted by linear regression (the exponential and linear model having the same number of parameters, the model with smallest sum of residuals was considered to be the best). In those cases, we used the following first model: $y = y_0 + \alpha \cdot T(T \in [12°, 25°])$.

And the nested model: $y = y_0 + \alpha \cdot T(T \in [12°, 25°]) + \beta(T - 25)(T \geq 25°C)$.

In all cases (temperature-dependent or temperature-independent), the thermal response of a feature was considered to change beyond the upper thermal limit if the *F*-test p-value was below 0.0014 (threshold 0.05 corrected for multiple testing 35 features, i.e., 0.05/35 = 0.0014; c.f. *Figure 1—source data 1*). If the p-value was above 0.0014, the feature's thermal response above the upper thermal limit was tagged as *unchanged* in *Figure 1—source data 1*.

We also performed Mann Whitney U-tests to test if two distributions were different (*Figure 4E–F*).

## Arrhenius kinetics—thermal response of cell cycle duration within the thermal range

Cell cycle durations within the thermal range were fitted to the following Arrhenius-like model (*Arrhenius, 1915*), to determine if the pace of cell division increased exponentially with temperature, as would be expected by Arrhenius kinetics:

$$duration = A \exp\left(\frac{-E}{k_B T}\right),$$

where $k_B$ is the Boltzmann constant, *T* is the temperature (in [K]), *E* is the activation energy describing the thermal dependence (in [eV]) and *A* is a normalization constant.

## Acknowledgements

Some strains were provided by the CGC, which is funded by NIH Office of Research Infrastructure Programs (P40 OD010440). We are very grateful to Alessandro De Simone for co-developing the pronuclei detection algorithm, to Johan Auwerx and Norman Moullan in his lab for guidance and technical help with the respiration experiments, as well as to Alexandra Bezler, Simon Blanchoud, Marie Delattre, Virginie Hamel, Andrew Hirst, Laurent Mouchiroud, Laurent Keller and Luc Pellerin for careful reading of the manuscript and useful comments. This work was supported by a SystemsX.ch Transition Postdoc Fellowship to AN (SXFSI0_141995). The authors declare no conflicts of interest.

## Additional information

### Funding

| Funder | Grant reference | Author |
|---|---|---|
| Swiss Initiative in Systems Biology | Transition Postdoc Fellowship (SXFSI0_141995) | Aitana Neves |

The funder had no role in study design, data collection and interpretation, or the decision to submit the work for publication.

### Author contributions

AN, Conception and design, Acquisition of data, Analysis and interpretation of data, Drafting or revising the article; CB, Contributed assistance for several experiments, including time-lapse recordings, progeny tests and respiration measurements; PG, Conception and design, Analysis and interpretation of data, Drafting or revising the article

## Additional files

### Supplementary file

• Source code 1. In-house scripts to analyze embryos (Matlab).

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
