## [Decision Letter]

[Editors’ note: this article was originally rejected after discussions between the reviewers, but the authors were invited to resubmit after an appeal against the decision.]

Thank you for choosing to send your work entitled “Cellular hallmarks reveal critical contribution of aerobic metabolism at the thermal limits” for consideration at *eLife*. Your full submission has been evaluated by Fiona Watt (Senior editor) and three peer reviewers, one of whom, Hans-Otto Portner, has agreed to reveal his identity. The decision was reached after discussions between the reviewers. Based on our discussions and the individual reviews below, we regret to inform you that your work will not be considered further for publication in *eLife*.

We are sending you the full comments of all three reviewers. In summary, they all agree that the work is very interesting. However, they have two key criticisms. First, the work does not specifically address the mechanistic link between changes in embryo size/shape, thermal effects and aerobic metabolism. Secondly, the presentation and discussion of the work is confusing. In discussing whether to reject the manuscript or encourage a revision, the reviewers concluded that it would take more than three months to generate the data to address the first criticism. If, however, you already have some additional mechanism that addresses the first point we would, of course, consider a rebuttal.

Reviewer #1:

Summary: This paper explores how embryos from two related nematode species respond to changes in temperature. The authors examine 35 features of 1-cell zygotes in response to temperature and identify those features that behave differently at the animals' thermal limits. In particular, they find that embryos tend to increase their surface to volume ratio at the thermal limits. Previous theoretical studies have suggested that changes in aerobic capacity underlie thermal stress. Consistent with this hypothesis, the authors found that depleting key metabolic enzymes also causes an increase in surface-to-volume ratio. Based on this correlation, the authors conclude that “aerobic metabolism is limiting at both lower and upper thermal limits”.

Critique:

1) The main thesis of this work is based on a correlation. There is no direct evidence that aerobic metabolism is limiting at the thermal limits. It would seem that simple experiments could be done to demonstrate this hypothesis more directly. For example, do embryos depleted for key aerobic metabolism enzymes become hypersensitive to temperature changes? Without additional experiments that test the hypothesis directly, the data are too preliminary.

2) The presentation of the data is often confusing. For example, Figure 1 reports on embryo size (do you mean volume?) and embryo shape (length/width). Surface-to-volume ratio, which in the Discussion seems to be the more relevant parameter, is not presented in the figure. As a result, it is very difficult to assess the significance of the differences/findings.

3) The data do not seem to always be fully explained in the text. The authors state that embryo size decreases below the lower thermal limit (12 to 10 degrees), but in fact embryo size increases dramatically (25% increase) between 14 to 12 degrees, before modestly going down between 12 and 10 degrees (Figure 2). This up-and-down behavior is not discussed.

4) The authors do not describe how the embryos were mounted on slides for imaging. If the embryos were placed under a coverslip, the pressure from the coverslip could affect their size and/or shape. If so, differences between different conditions could reflect differences in the resistance of the eggshell to the coverslip pressure, not actual differences in size.

Reviewer #2:

This is an interesting paper relating observations on functional changes to thermal limitation in *Caenorhabditis elegans* embryos. This non-conventional way of addressing this issue will be stimulating for the animal biology community. Nonetheless the text should be amended and jargon reduced to make it more widely accessible. Some issues may be commonly expressed that way in the nematode world, but deserve explanation, e.g. why and how does surface to volume ratio relate to aerobic metabolism?

This writing, in the Introduction, is a bit imprecise: the biogeographic range relates to the thermal window of a species, so both upper and lower limits. This fundamental thermal range is not fully exploited but forms the basis of the exploited range.

Results and Discussion, third paragraph: it is not correct that the onset of anaerobic metabolism characterizes the first line of thermal limitation. The cited papers by Pörtner and colleagues refer to what is called the concept of oxygen and capacity limited thermal tolerance (OCLTT, cf. Pörtner 2010 J. exp Biol.). The earliest limit is a loss in performance caused by a mismatch in oxygen supply capacity and associated costs as well as hypoxemia constraining performance capacity, leaving less energy for e.g. growth (Pörtner and Knust, 2007, Science). So thermal limits then lead to the onset of anaerobic metabolism. These findings have recently been set into perspective across organism domains (Storch et al., 2014, Global Change Biology), which should caution authors with respect to some of their statements on prokaryotes and get their rationale into clearer shape.

At the end of the Introduction: MASROS seems as a spinoff of OCLTT, this link should be mentioned.

In the embryos oxygen supply may become constrained by diffusion limitations of supply and demand.

Results and Discussion, seventh paragraph: the simulation of the effect of hypoxemia on aerobic scope by lowering ATP contents is a great trick to address the question. I was missing estimates of change in ATP levels? Are other effects conceivable? Under which oxygen supply conditions would ATP concentrations change in vivo?

Results and Discussion, fourth paragraph: you should not talk about thermal limitation before actually identifying them. Is the plateauing of cell cycle an early limitation (e.g. pejus limits?)

Results and Discussion, sixth paragraph: ditto, you should use all of these data later to fit it into a conceptual framework of limitation.

This leaves my criticism on the gaps in the interpretation of these interesting data. Studying the OCLTT concept may provide the framework and terminology needed to build a coherent story, considering various levels of thermal limitation. This may also lead to a fresh look at some of their data and on how to best integrate them.

Reviewer #3:

Although this study provides novel quantitative observations on the thermal sensitivity of early embryonic development, the key conclusions of the paper are only weakly supported by the experimental data. Moreover, data presentation and interpretation as well as the placing of the observations in their general context are often unclear.

1) The link between temperature effects on embryo size/shape and aerobic metabolism is not explicitly tested. The experimental data (RNAi) is suggestive but not a single experiment (or literature citation) is provided that would inform about temperature-dependent changes in aerobic metabolism in embryos. Therefore, while the RNAi data is suggestive, they are clearly too preliminary to support the strong conclusions made.

2) The actual data is not sufficiently discussed. For example, a number of phenotypes measured show non-linear temperature responses, including reduced embryo size at lower temperatures (e.g. Figures 2 and 3). In *C. elegans*, embryo size seems reduced only when comparing 10C to 12C treatment; in contrast, when comparing all temperatures above 12C to 10C, embryo size seems increased. These non-linear responses are in my view incongruent with the hypothesis on aerobic metabolism—yet these experimental data are not discussed in detail.

3) The thermal limits are defined as upper and lower temperatures at which embryonic hatching is reduced to ∼90% but it is unclear what leads to this 10% reduction, e.g. if changes in early embryonic development contribute to this. Moreover, for data obtained beyond thermal limits it is unclear if any of the changes observed (e.g. embryo shape) contribute to increased embryonic mortality. [Also: thermal range is not a generic term but should be explicitly linked to the phenotype measured, e.g. embryonic viability, fertility, survival. A given genotype will have different thermal limits for these different phenotypes].

4) The data focuses on mean changes in developmental phenotypes although observed temperature effects (as many stressors in general) strongly increase variance of phenotypes at extreme temperatures (e.g. debuffering). This is not analysed or discussed appropriately.

5) The evolutionary context provided for this study is a bit naïve, sometimes wrong (e.g. first two sentences of the Introduction). For example, thermal adaptation is rarely a species-specific trait, but there is ample genetic variation in thermal responses within a species. This is important also in the context of this study: *C. briggsae* shows extensive genetic variation in upper thermal limits of reproductive capacity with genetically distinct tropical and temperate clades (29). [In addition, *C. briggsae* occurs in many cold regions, world-wide, cf. Results and Discussion, last line of first paragraph]. Similarly, the discussion of literature on temperature effects on aerobic metabolism etc. is not sufficiently clear to follow the authors' logic.

[Editors’ note: what now follows is the decision letter after the authors submitted for further consideration.]

Thank you for sending your work entitled “Cellular hallmarks reveal critical contribution of aerobic metabolism at the thermal limits” for consideration at *eLife*. Your rebuttal was evaluated by Fiona Watt (Senior editor) and the three original reviewers.

We feel that your manuscript can potentially be published in *eLife*, but only subject to your addressing the remaining concerns of Reviewers 1 and 3. If you choose to submit a revised version of the manuscript it will be your final opportunity to satisfy the reviewers—we are not willing to consider any further rounds of revision. This reviewers’ major criticisms concern the RNAi experiments and are described in lightly edited format below:

Overall, the manuscript presents many measurements (e.g. size, shape, viability) but how these integrate and how they relate to aerobic metabolism and thermal limits remain mostly unclear. No direct respiration rate measurements were taken in RNAi-treated animals and it remains unclear to what extent aerobic metabolism and embryonic lethality are causally linked in *wt* animals. Statistical analyses for the new data appear lacking and the experimental design requires more explanation. The authors show that embryos at the thermal limit are less sensitive to depletion of mitochondrial activity by RNAi, than animals in the thermal range. At the upper limit of the thermal range, the data show that RNAi depletion of two mitochondrial enzymes (but curiously not a third one) has no effect on viability. There are no controls to show that the RNAi depletion works equally well at all temperatures and for all three genes, so it is possible that this odd behavior is due to incomplete depletion at different temperatures and/or for different genes. In addition, the logic used to interpret these experiments is not clear. It would seem that if respiration becomes limiting at the thermal limit, then embryos near the limit should become more sensitive, not less, to depletion of respiratory components. In this line of reasoning, the new data would suggest that, in fact, respiration is not the rate limiting step at the thermal limit.

In addition, reviewer 1 was still troubled by the possibility that changes in eggshell composition in embryos raised at different temperatures could cause embryos to appear larger or smaller due to different resilience to the pressure of the cover slip used for imaging. We would be grateful if you could deal with this possibility when discussing your data.

---

## [Author Response]

[Editors’ note: the author responses to the first round of peer review follow.]

After carefully reading the comments and suggestions made by the reviewers, we conducted experiments and modified the manuscript to address in full the issues raised by the three reviewers, as described in detail in the accompanying rebuttal letter. In particular, the first criticism raised by reviewers #1 and #3 concerning the lack of direct evidence for aerobic metabolism being restricted at the thermal limits has now been addressed. As we show in the new Figure 4, respiration measurements reveal that aerobic metabolism is indeed reduced beyond both lower and upper thermal limits. Moreover, we also uncovered that embryos depleted of distinct components of the mitochondrial respiratory chain are less affected towards the thermal limits than within the thermal range (new Figure 4). With these new findings, our work provides for the first time compelling evidence that restricted aerobic metabolism is a key feature of thermal limits in a terrestrial metazoan organism. Moreover, we altered the text to render it more accessible to a broader audience, clarifying the presentation and discussion of the work, thus addressing the second main criticism of the reviewers.

We are sending you the full comments of all three reviewers. In summary, they all agree that the work is very interesting. However, they have two key criticisms. First, the work does not specifically address the mechanistic link between changes in embryo size/shape, thermal effects and aerobic metabolism. Secondly, the presentation and discussion of the work is confusing. In discussing whether to reject the manuscript or encourage a revision, the reviewers concluded that it would take more than three months to generate the data to address the first criticism. If, however, you already have some additional mechanism that addresses the first point we would, of course, consider a rebuttal.

*Reviewer #1*:

*Summary*: *This paper explores how embryos from two related nematode species respond to changes in temperature. The authors examine 35 features of 1-cell zygotes in response to temperature and identify those features that behave differently at the animals' thermal limits. In particular, they find that embryos tend to increase their surface to volume ratio at the thermal limits. Previous theoretical studies have suggested that changes in aerobic capacity underlie thermal stress. Consistent with this hypothesis, the authors found that depleting key metabolic enzymes also causes an increase in surface-to-volume ratio. Based on this correlation, the authors conclude that “aerobic metabolism is limiting at both lower and upper thermal limits”.*

*Critique*:

*1) The main thesis of this work is based on a correlation. There is no direct evidence that aerobic metabolism is limiting at the thermal limits. It would seem that simple experiments could be done to demonstrate this hypothesis more directly. For example, do embryos depleted for key aerobic metabolism enzymes become hypersensitive to temperature changes? Without additional experiments that test the hypothesis directly, the data are too preliminary*.

We thank the reviewer for raising this very important point. Indeed, our initial submission was missing direct evidence that aerobic metabolism is restricted at the thermal limits. This critical issue has now been addressed by measuring the respiration rate of wild-type *C. elegans* embryos from 9°C to 28°C. As we now show in Figure 4, this novel experiment revealed that respiration increases exponentially within the thermal range, as predicted by Arrhenius-like kinetics (5). Strikingly in addition, we found that respiration decreases both below the lower thermal limit and above the upper thermal limit. These findings indicate that aerobic capacity is indeed reduced beyond both thermal limits as compared to within the thermal range.

Importantly in addition, as suggested by the reviewer, we further challenged our findings by now measuring embryonic viability over a wide range of temperatures upon depletion of key aerobic metabolism proteins (subunits of complex I, III and V of the mitochondrial respiratory chain). We reasoned that if aerobic capacity is utilized to a lesser extent for energy production beyond both thermal limits than within the thermal range, then compromising mitochondrial activity should also have less of an impact at the thermal limits than within the thermal range. Remarkably, we indeed found that embryonic lethality was reduced towards the thermal limits as compared to within the thermal range in *atp-2(RNAi), cyc-1(RNAi)* and *nuo-1(RNAi)* embryos (Figure 4). These results offer strong experimental support to the notion that the capacity of the mitochondrial respiratory chain is indeed reduced beyond both thermal limits.

All these critical findings are now presented and discussed in our manuscript in the Results section entitled “Cellular hallmarks of the thermal limits are recapitulated when impairing aerobic metabolism”*.*

*2) The presentation of the data is often confusing. For example,*
Figure 1
*reports on embryo size (do you mean volume?) and embryo shape (length/width). Surface-to-volume ratio, which in the Discussion seems to be the more relevant parameter, is not presented in the figure. As a result, it is very difficult to assess the significance of the differences/findings*.

We apologize for having been insufficiently clear in our presentation of the data. We have put great effort in extensively rewriting the manuscript in order to present the data more clearly. In Figures 2 and 3, we report embryo area, which we use as a proxy of embryo size given the rotational symmetry of *C. elegans* embryos (see Materials and methods). In the previous version of the manuscript, surface-to-volume ratios were not reported because they were merely estimated and not measured, as explained in the Materials and methods of the initial submission. In the new version of the manuscript, we reduce the importance of surface-to-volume ratios and focus instead on the signature cellular changes at the thermal limits, namely embryo size reduction below the lower thermal limit and embryo elongation above the upper thermal limit, both of which have been directly measured. See Results section entitled “Cellular hallmarks of the thermal limits are recapitulated when impairing aerobic metabolism”.

*3) The data do not seem to always be fully explained in the text. The authors state that embryo size decreases below the lower thermal limit (12 to 10 degrees), but in fact embryo size increases dramatically (25% increase) between 14 to 12 degrees, before modestly going down between 12 and 10 degrees (*Figure 2*). This up-and-down behavior is not discussed*.

We thank the reviewer for having brought up this important point. As we now explain better (see Introduction, third paragraph and Results section “Embryo size and shape are sensitive to the thermal limits*”*), the fact that body size is smaller at higher temperatures within the thermal range is common among ectotherms and has been coined the temperature-size rule (6; 17). Such a size decrease with warmer temperatures is believed to be an adaptive response to preserve aerobic capacity by yielding an increase in surface-to-volume ratio and thus potentially in oxygen supply by facilitated diffusion (8).

In the previous version of the manuscript, we had focused our attention solely on the reversal of this temperature-size rule below the lower thermal limit, which has been reported previously only in protists and in *Drosophila* (40; 7), and hypothesized to be driven by restricted aerobic metabolism below the lower thermal limit. The new version of the manuscript maintains this focus, while putting it in the context of the temperature-size rule delineated above, thus clarifying what may have appeared as an unexpected behavior at the lower thermal limit.

*4) The authors do not describe how the embryos were mounted on slides for imaging. If the embryos were placed under a coverslip, the pressure from the coverslip could affect their size and/or shape. If so, differences between different conditions could reflect differences in the resistance of the eggshell to the coverslip pressure, not actual differences in size*.

This point indeed deserved some further clarification. As we now explain in the first paragraph of Materials and methods, embryos were dissected in 1x M9 medium tempered at the culture temperature, mounted on slides, placed under a coverslip and imaged using time-lapse DIC microscopy. Considering the crowded compressive environment of the uterus in the intact animal, and considering furthermore that the same mounting procedure was followed for all specimens at all temperatures, we surmise that the observed alterations in thermal response of embryo size and shape at given temperatures are not due to the mounting procedure.

*Reviewer #2*:

*This is an interesting paper relating observations on functional changes to thermal limitation in* Caenorhabditis elegans *embryos*. *This non-conventional way of addressing this issue will be stimulating for the animal biology community. Nonetheless the text should be amended and jargon reduced to make it more widely accessible. Some issues may be commonly expressed that way in the nematode world, but deserve explanation, e.g. why and how does surface to volume ratio relate to aerobic metabolism?*

We thank the reviewer for acknowledging our non-conventional way of addressing this issue and also believe that our findings will be stimulating for the animal biology community at large.

*This writing, in the Introduction, is a bit imprecise: the biogeographic range relates to the thermal window of a species, so both upper and lower limits. This fundamental thermal range is not fully exploited but forms the basis of the exploited range*.

We agree with the reviewer that our writing was somewhat imprecise and now clearly mention in the Introduction that other factors also determine the actual exploited range:

“Partly as a result, organisms tend to distribute in the ocean and on land according to latitude as well as depth and altitude, although other elements such as availability of food and light also play a role in shaping preferred habitats (28; 27; 29)*.”*

*Results and Discussion, third paragraph: it is not correct that the onset of anaerobic metabolism characterizes the first line of thermal limitation. The cited papers by Pörtner and colleagues refer to what is called the concept of oxygen and capacity limited thermal tolerance (OCLTT, cf. Pörtner 2010 J. exp Biol.). The earliest limit is a loss in performance caused by a mismatch in oxygen supply capacity and associated costs as well as hypoxemia constraining performance capacity, leaving less energy for e.g. growth (Pörtner and Knust, 2007, Science). So thermal limits then lead to the onset of anaerobic metabolism. These findings have recently been set into perspective across organism domains (Storch et al., 2014, Global Change Biology), which should caution authors with respect to some of their statements on prokaryotes and get their rationale into clearer shape*.

We acknowledge that our writing was far too simplistic in an attempt to accommodate a broad readership, thus loosing part of its meaning. We have now made sure to clearly mention that a mismatch in oxygen supply and demand determines a first thermal limit (27; 28). We now also explicitly describe the experiments in both aquatic organisms and prokaryotes to better guide the reader through what has been achieved so far and what still needed further validation prior to our work. See Introduction:

*“*Oxygen supply has been postulated to play a role in setting thermal limits in multicellular organisms. […] even in the prokaryote *E. coli* (Morrison and Shain, 2008), suggesting that energy limitation may be a general feature that characterizes life on the edge of the thermal range*.”*

*At the end of the Introduction: MASROS seems as a spinoff of OCLTT, this link should be mentioned*.

Indeed, MASROS is based on the OCLTT, and we now first explain OCLTT in the Introduction, and only then describe how this yields to the MASROS. See Introduction:

“Body size decreases with augmented temperature in the vast majority of ectotherms (“temperature-size rule”) (6; 17), thereby increasing surface to volume ratio and thus potentially oxygen availability. (...) This has led to the suggestion that alterations in cell size in response to changes in temperature within the thermal range are adaptive responses to preserve aerobic capacity, which has been dubbed the MASROS hypothesis (Maintain Aerobic Scope—Regulate Oxygen Supply) (8).”

*In the embryos oxygen supply may become constrained by diffusion limitations of supply and demand*.

Results and Discussion, seventh paragraph: the simulation of the effect of hypoxemia on aerobic scope by lowering ATP contents is a great trick to address the question. I was missing estimates of change in ATP levels? Are other effects conceivable? Under which oxygen supply conditions would ATP concentrations change in vivo?

This is indeed an important point. Measurements of ATP levels have already been performed on extracts from adult worms, revealing a ∼2-5 fold decrease in ATP levels in conditions that compromise the mitochondrial respiratory chain (15). We sought to perform analogous measurements on embryonic extracts, but observed a confounding lack of reproducibility. This may stem in part from the fact that extracts are by necessity from unsynchronized embryos (as large scale synchronization is not possible in this system) and that ATP levels may vary considerably between the early and the late stages of embryogenesis.

As we now discuss in the Results and Discussion, other effects are indeed conceivable, including changes in cytosolic pH or an increase in reactive oxygen species.

To our knowledge, it is not known under which oxygen conditions ATP levels would change in vivo. Although this is an interesting point that will deserve further investigation, we are of the view that it falls outside of the central scope of our manuscript.

Results and Discussion, fourth paragraph: you should not talk about thermal limitation before actually identifying them. Is the plateauing of cell cycle an early limitation (e.g. pejus limits?)

*Results and Discussion, sixth paragraph: ditto, you should use all of these data later to fit it into a conceptual framework of limitation*.

We define the thermal limits as the edges of the thermal range within which >90% of embryos hatch. Considering the new respiration measurements presented in Figure 4, we believe that these limit temperatures are very similar to the pejus limits defined by others (27).

Throughout the text, we then use the thermal limits defined by embryonic viability tests and examine various cellular features at those limits. Interestingly, we uncovered that the thermal response of select cellular features changed precisely at the limit temperatures defined by the embryonic viability tests (see Figure 1). This is now mentioned in our manuscript in the Results section.

*This leaves my criticism on the gaps in the interpretation of these interesting data. Studying the OCLTT concept may provide the framework and terminology needed to build a coherent story, considering various levels of thermal limitation. This may also lead to a fresh look at some of their data and on how to best integrate them*.

We warmly thank the reviewer for encouraging us in this direction. As a result, we rewrote the whole manuscript using the framework and terminology from the OCLTT and MASROS hypotheses, and believe that the text is now more coherent and the data and interpretations more solid.

*Reviewer #3*:

*Although this study provides novel quantitative observations on the thermal sensitivity of early embryonic development, the key conclusions of the paper are only weakly supported by the experimental data. Moreover, data presentation and interpretation as well as the placing of the observations in their general context are often unclear*.

We agree with the reviewer that the key conclusions drawn from the experiments presented in the initial submission required further validation. As we mentioned earlier in this rebuttal (see response to point 1 from reviewer #1), we conducted both respiration measurements as well as progeny tests in embryos with impaired aerobic metabolism over a wide range of temperatures. The resulting data strongly support the notion that aerobic metabolism is indeed restricted beyond the thermal limits. We also ensured that the data is clearly presented and that our interpretations are thoroughly discussed throughout the new rendition of the manuscript.

*1) The link between temperature effects on embryo size/shape and aerobic metabolism is not explicitly tested. The experimental data (RNAi) is suggestive but not a single experiment (or literature citation) is provided that would inform about temperature-dependent changes in aerobic metabolism in embryos. Therefore, while the RNAi data is suggestive, they are clearly too preliminary to support the strong conclusions made*.

We thank the reviewer for suggesting this crucial experiment, which was also suggested by reviewer #1. We kindly ask reviewer #3 to refer to the answer to point 1 of reviewer #1, where we explain in detail how this issue has been addressed.

*2) The actual data is not sufficiently discussed. For example, a number of phenotypes measured show non-linear temperature responses, including reduced embryo size at lower temperatures (e.g.*
Figures 2 and 3*). In* C. elegans, *embryo size seems reduced only when comparing 10C to 12C treatment; in contrast, when comparing all temperatures above 12C to 10C, embryo size seems increased. These non-linear responses are in my view incongruent with the hypothesis on aerobic metabolism—yet these experimental data are not discussed in detail*.

We acknowledge that the data within the thermal range, in particular regarding embryo size, was not discussed sufficiently in the initial submission. Reviewer #1 had a similar comment, which we addressed above (see answer to her/his point 3). As we explain in that paragraph, such non-linear responses are actually very supportive of the hypothesis on aerobic metabolism.

*3) The thermal limits are defined as upper and lower temperatures at which embryonic hatching is reduced to ∼90% but it is unclear what leads to this 10% reduction, e.g. if changes in early embryonic development contribute to this. Moreover, for data obtained beyond thermal limits it is unclear if any of the changes observed (e.g. embryo shape) contribute to increased embryonic mortality*.

This point is well taken: we indeed do not know which stage of embryonic development is most sensitive to the thermal limits and also cannot ascertain whether the changes that we identified during the first cell cycle contribute to the observed lethality. We now explicitly state this limitation in our Conclusions. Interestingly, however, we uncovered that the thermal response of select cellular features changed precisely at the limit temperatures defined by embryonic viability tests, suggesting at the minimum strong correlation.

*[Also: thermal range is not a generic term but should be explicitly linked to the phenotype measured, e.g. embryonic viability, fertility, survival. A given genotype will have different thermal limits for these different phenotypes]*.

We fully agree with the reviewer that thermal range is not a generic term but should always be linked to some phenotype. In our manuscript, we have linked it to embryonic viability. After defining this range at the beginning of the Results section, we then asked if cellular changes were readily observable at those limit temperatures defined by embryonic viability tests, and strikingly found this to be the case for some (see Figures 2 and 3).

4) The data focuses on mean changes in developmental phenotypes although observed temperature effects (as many stressors in general) strongly increase variance of phenotypes at extreme temperatures (e.g. debuffering). This is not analysed or discussed appropriately.

We thank the reviewer for highlighting this point. Actually, we represented the data in terms of boxplots in order to make sure that the average, the variance and any potential skew in the data would be clearly visible. We did not discuss this in our manuscript, because we had verified as prior to submission that the variance of cellular features did not increase beyond the thermal limits as compared to within the thermal range. This can be seen also from the colored boxes in Figures 2 and 3 representing data beyond the thermal limits. This is now mentioned in the legend of Figure 2.

*5) The evolutionary context provided for this study is a bit naïve, sometimes wrong (e.g. first two sentences of the Introduction). For example, thermal adaptation is rarely a species-specific trait, but there is ample genetic variation in thermal responses within a species. This is important also in the context of this study:* C. briggsae *shows extensive genetic variation in upper thermal limits of reproductive capacity with genetically distinct tropical and temperate clades (*[29]*). [In addition*, C. briggsae *occurs in many cold regions, world-wide, cf. Results and Discussion, last line of first paragraph]*.

We agree with the reviewer that the evolutionary context initially provided was somewhat naïve. We fully acknowledge that thermal adaptation is generally not a species-specific trait and that the introductory sentences in our initial submission were overly simplistic. We have now rephrased the first paragraph of our Introduction in order to better account for the multiple factors that may determine an organism's thermal range. It now reads:

“All organisms live within a given thermal range, beyond which growth and fecundity decrease (28). Partly as a result, organisms tend to distribute in the ocean and on land according to latitude as well as depth and altitude, although other elements such as availability of food and light also play a role in shaping preferred habitats (28; 27; 29). Despite their importance, the mechanisms that set the thermal limits remain incompletely understood*.”*

We thank the reviewer for noting that *C. briggsae* also occurs in many cold regions and, therefore, that our writing was too generic. We have slightly modified the text as follows to address this point (Results and Discussion):

*“*We thus found that the thermal limits of *C. elegans* were of 12° and 25°C (Figure 1), and those of *C. briggsae* of 14° and 27°C (Figure 1), in line with the fact that *C. briggsae* usually lives in warmer climates than *C. elegans* (29)*.”*

*Similarly, the discussion of literature on temperature effects on aerobic metabolism etc. is not sufficiently clear to follow the authors' logic*.

We apologize for the confusion generated by the previous version of the manuscript and sincerely hope that our logic is now easier to follow with this new version. We kindly ask the reviewer to refer to our new Introduction and Results sections.

[Editors’ note: the author responses to the re-review follow.]

Overall, the manuscript presents many measurements (e.g. size, shape, viability) but how these integrate and how they relate to aerobic metabolism and thermal limits remain mostly unclear.

We summarize below how, in our view, our findings form a coherent story. We have also further clarified this coherence throughout the revised text of the manuscript.

1) Changes in embryo size and shape are observed beyond both thermal limits in embryos of *C.* elegans and *C. briggsae*.

2) Aerobic metabolism, measured as respiration, is reduced beyond both thermal limits in *C. elegans* embryos.

3) In order to test whether the observed changes in size and shape beyond thermal limits result from reduced aerobic capacity (i.e. whether point 1 above results from point 2 above), we analyzed cellular features within the thermal range in embryos compromised for mitochondrial respiratory chain function. We find that both embryo size and shape are altered under these conditions, providing support for the hypothesis that such changes beyond the thermal limits in the wild-type indeed reflect reduced aerobic metabolism.

4) The respiration experiments establish that aerobic metabolism is reduced at the thermal limits. This piece of data alone is insufficient to distinguish between a scenario in which the energetic needs of the embryo are not satisfied due to insufficient aerobic metabolism and one in which these needs are actually fulfilled to some extent despite reduced respiration, perhaps because other metabolic routes are used to a larger extent. Interestingly, these two scenarios predict a different outcome in embryos compromised for mitochondrial respiratory chain function. If aerobic metabolism became insufficient beyond the thermal limits, then further compromising mitochondrial activity should have more of an impact at the thermal limits than within the thermal range. By contrast, if energetic needs could be fulfilled at the least to some extent despite reduced respiration beyond the thermal limits, then further compromising mitochondrial activity should have less of an impact at the thermal limits than within the thermal range. Importantly, we uncovered that the latter is the case, demonstrating that aerobic metabolism is restricted towards the thermal limits, as reported in Figure 4. We have rewritten in an extensive manner the corresponding section in the revised manuscript to further clarify this important point (Results and Discussion section, subsection headed “Cellular hallmarks of the thermal limits are recapitulated when impairing aerobic metabolism”).

No direct respiration rate measurements were taken in RNAi-treated animals.

Following the suggestion of the reviewer, we set out to verify that respiration is decreased in embryos with compromised mitochondrial respiratory chain function. Since ATP-2 and NUO-1 are part of complex V and I, respectively, respiration might still be observed upon their depletion despite mitochondrial respiratory chain being compromised (see [12]), so that respiration measurements would not necessarily be telling in these cases Instead, respiration should be decreased upon compromising complex III function, as in *cyc-1(RNAi)* embryos. As anticipated, we found indeed that respiration is markedly diminished in such embryos, being on average 56% ± 13% of the wild-type embryonic respiration levels in these experimental conditions (t-test p-value < 10^-3^)*.* These new results are reported in the Results and Discussion section, under “Cellular hallmarks of the thermal limits are recapitulated when impairing aerobic metabolism” (see also updated corresponding Materials and methods*,* subsection headed “Measurement of embryonic respiration”).

*It remains unclear to what extent aerobic metabolism and embryonic lethality are causally linked in* wt *animals.*

This point is well taken and we now explicitly state this limitation in the Results and Discussion (in the subsection headed “Cellular hallmarks of the thermal limits are recapitulated when impairing aerobic metabolism”).

Statistical analyses for the new data appear lacking and the experimental design requires more explanation.

We thank the reviewer for having spotted this omission; apologies about this. We had in fact conducted an F-test analysis (as described in Supplementary file 1 for other measurements) and have now added the corresponding p-values in the subsection “Cellular hallmarks of the thermal limits are recapitulated when impairing aerobic metabolism” (F-test p-value < 10^-4^ and F-test p-value < 10^-10^ below the lower and above the upper thermal limit, respectively).

We have also expanded the text in the Materials and methods section in order to explain better the experimental design of the respiration experiments.

The authors show that embryos at the thermal limit are less sensitive to depletion of mitochondrial activity by RNAi, than animals in the thermal range. At the upper limit of the thermal range, the data show that RNAi depletion of two mitochondrial enzymes (but curiously not a third one) has no effect on viability. There are no controls to show that the RNAi depletion works equally well at all temperatures and for all three genes, so it is possible that this odd behavior is due to incomplete depletion at different temperatures and/or for different genes.

In order to address this potential issue, we performed progeny tests at different temperatures as a proxy for depletion efficiency using RNAi against AIR-1, a serine/threonine kinase required for spindle assembly (21), a process not known to be related to metabolic status. As we show below (see panel A), we found that *air-1(RNAi)* was 100% embryonic lethal at 12°C, 20°C and 24°C, as anticipated from previous work (21). In order to titrate the phenotype, we performed double RNAi by mixing bacteria expressing dsRNA against *air-1* with bacteria expressing dsRNA against *gfp* in a 1:3 ratio. Importantly, we found in this case that lethality was greater at 12°C and at 24°C than at 20°C (see panel B). Hence, the RNAi phenotype is actually stronger for *air-1* towards the lower and the upper thermal limit than at 20°C, indicating that the results we uncovered when targeting mitochondrial respiratory chain components (Figure 4 of the manuscript) are not due to a general temperature-dependent response of RNAi. These results are shown in Figure 4—figure supplement 2 and reported in the Materials and methods section.

In addition, the logic used to interpret these experiments is not clear. It would seem that if respiration becomes limiting at the thermal limit, then embryos near the limit should become more sensitive, not less, to depletion of respiratory components. In this line of reasoning, the new data would suggest that, in fact, respiration is not the rate limiting step at the thermal limit.

We agree with the reviewer that aerobic metabolism is not limiting at the thermal limits (as we wrote in our very initial submission), and apologize for probably not having been sufficiently clear on this point in our last submission. As we explain in point 4 above, our data supports a mechanism whereby aerobic metabolism is restricted (and not limited) at the thermal limits. As mentioned also above, we have extensively modified the main text to make this clearer (Results and Discussion section, in the subsection headed “Cellular hallmarks of the thermal limits are recapitulated when impairing aerobic metabolism”).

In addition, reviewer 1 was still troubled by the possibility that changes in eggshell composition in embryos raised at different temperatures could cause embryos to appear larger or smaller due to different resilience to the pressure of the cover slip used for imaging. We would be grateful if you could deal with this possibility when discussing your data.

We have updated the Materials and methods to explicitly raise this possibility.